# Structural and dynamic impacts of single-atom disruptions to guide RNA interactions within the recognition lobe of *Geobacillus stearothermophilus* Cas9

**Helen B Belato[1†], Alexa L Knight[1†], Alexandra M D'Ordine[1], Chinmai Pindi[2], Zhiqiang Fan[3], Jinping Luo[3], Giulia Palermo[2], Gerwald Jogl[1,4], George P Lisi[1,4]***

[1]Department of Molecular Biology, Cell Biology and Biochemistry, Brown University, Providence, United States; [2]Departments of Bioengineering and Chemistry, University of California, Riverside, Riverside, United States; [3]Brown University Transgenic Mouse and Gene Targeting Facility, Providence, United States; [4]Brown University RNA Center, Providence, United States

**\*For correspondence:**
george_lisi@brown.edu

[†]These authors contributed equally to this work

**Competing interest:** The authors declare that no competing interests exist.

## eLife Assessment

This study offers **valuable** insights into the conformational dynamics of the nucleic acid recognition lobe of GeoCas9, a thermophilic Cas9 from Geobacillus stearothermophilus. The authors investigate the influence of local dynamics and allosteric regulation on guide RNA binding affinity and DNA cleavage specificity through molecular dynamics simulations, advanced NMR techniques, RNA binding studies, and mutagenesis. While the mutations studied do not lead to significant changes in GeoCas9 cleavage activity, the study provides **convincing** evidence for the role of allosteric mechanisms and interdomain communication in Cas9 enzymes, and will be of great interest to biochemists and biophysicists exploring these complex systems.

**Abstract** The intuitive manipulation of specific amino acids to alter the activity or specificity of CRISPR-Cas9 has been a topic of great interest. As a large multi-domain RNA-guided endonuclease, the intricate molecular crosstalk within the Cas9 protein hinges on its conformational dynamics, but a comprehensive understanding of the extent and timescale of the motions that drive its allosteric function and association with nucleic acids remains elusive. Here, we investigated the structure and multi-timescale molecular motions of the recognition (Rec) lobe of *Geo*Cas9, a thermophilic Cas9 from *Geobacillus stearothermophilus*. Our results provide new atomic details about the *Geo*Rec subdomains (*Geo*Rec1, *Geo*Rec2) and the full-length domain in solution. Two rationally designed mutants, K267E and R332A, enhanced and redistributed micro-millisecond flexibility throughout *Geo*Rec, and NMR studies of the interaction between *Geo*Rec and its guide RNA showed that mutations reduced this affinity and the stability of the ribonucleoprotein complex. Despite measured biophysical differences due to the mutations, DNA cleavage assays reveal no functional differences in on-target activity, and similar specificity. These data suggest that guide RNA interactions can be tuned at the biophysical level in the absence of major functional losses but also raise questions about the underlying mechanism of *Geo*Cas9, since analogous single-point mutations have significantly impacted on- and off-target DNA editing in mesophilic *Streptococcus pyogenes* Cas9. A K267E/R332A double mutant did also did not enhance *Geo*Cas9 specificity, highlighting the robust tolerance of mutations to the Rec lobe of *Geo*Cas9 and species-dependent complexity of Rec across Cas9 paralogs. Ultimately, this work provides an avenue by which to modulate the structure, motion,

and guide RNA interactions at the level of the Rec lobe of *Geo*Cas9, setting the stage for future studies of *Geo*Cas9 variants and their effect on its allosteric mechanism.

## Introduction

The vast majority of Cas systems explored as genome editors originate from mesophilic hosts. The emergence of the thermophilic *Geo*Cas9, with DNA cleavage function up to 85 °C, can expand CRISPR technology to higher temperature regimes and stabilities (*Belato et al., 2022b*; *Harrington et al., 2017*), but its regulatory mechanism relative to canonical Cas9s must be established. The *Spy*Cas9, which originates from the mesophilic *Streptococcus pyogenes*, as well as *Geo*Cas9, are both effectors of Type-II CRISPR systems. Interestingly, the Type II-A *Spy*Cas9 has been by far the most used Cas enzyme, including in ongoing clinical trials (*Zhang et al., 2023*; *Li et al., 2023*). But Cas9 homologs of the Type II-C class, such as *Neisseria meningitis* (*Nme*Cas9) and *Campylobacter jejuni* (*Cje*Cas9), to which *Geo*Cas9 belongs, have been validated for mammalian genome editing (*Harrington et al., 2017*; *Kim et al., 2017*; *Lee et al., 2016*), reinforcing the need to better understand this CRISPR class.

The similar domain arrangements of *Geo*Cas9 and *Spy*Cas9 led us to initially speculate that these could share atomic level mechanistic similarities (*Belato et al., 2022a*). *Geo*Cas9 utilizes a guide RNA (gRNA) to localize and unwind a double-stranded DNA (dsDNA) target after recognition of its 5'NNNNCRAA-3' protospacer adjacent motif (PAM; *Harrington et al., 2017*; *Jinek et al., 2012*). Upon recognition of the PAM sequence by the PAM-Interacting (PI) domain, Cas9-bound guide (gRNA) forms an RNA:DNA hybrid with the target DNA strand. Initially thought to be part of the PI domain (*Harrington et al., 2017*), the wedge (WED) domain recognizes the repeat:anti-repeat region of the gRNA and the dsDNA upstream of the target region (*Eggers et al., 2024*). The Rec lobe of Cas9 is responsible for orienting the RNA:DNA hybrid, as well as the adjacent nuclease domains, into their active conformations (*Palermo et al., 2018*; *Dagdas et al., 2017*; *Mir et al., 2018*; *Jiang et al., 2015*). Coordinated cleavage of the target and non-target DNA strand then occurs via the HNH and RuvC nucleases, respectively. The *Geo*Cas9 nuclease active sites within HNH and RuvC are spatially distinct from the PAM recognition site in the PI domain, necessitating structural and dynamic changes that allosterically couple dsDNA binding to cleavage. Biochemical (*Dagdas et al., 2017*; *Sternberg et al., 2015*; *Chen et al., 2017*) and structural (*Skeens et al., 2024*; *East et al., 2020*) experiments using the extensively studied *Spy*Cas9 have revealed that its function is governed by a sophisticated allosteric mechanism that transfers gRNA and dsDNA binding information from the Rec lobe to the distal catalytic sites. A dynamically driven allosteric signal spans the HNH domain of *Spy*Cas9, enabled by the plasticity of the Rec lobe, which orchestrates the conformational activation required for DNA cleavage (*Sternberg et al., 2015*; *Palermo et al., 2016*). Our prior work revealed a divergence in the timescales of allosteric motions in the *Spy*Cas9 and *Geo*Cas9 HNH domains (*Belato et al., 2022a*; *East et al., 2020*) suggesting an unusually flexible HNH and unique allosteric mode of regulation for *Geo*Cas9. It is therefore also possible that docking of the gRNA with *Geo*Cas9, and thus its interaction with the RNA:DNA hybrid, may differ from the *Spy*Cas9 system, as *Geo*Cas9 contains a truncated Rec lobe with only two of the three canonical subdomains.

The high thermal stability and more compact size of *Geo*Cas9 (it is 281 residues shorter than *Spy*Cas9) can be especially important for in vivo delivery applications, since promising viral vectors (i.e. adeno-associated virus, AAV) have cargo capacities of ~4.7 kb (*Wu et al., 2010*), which prevents *Spy*Cas9-gRNA packaging into a single AAV vector but permits 'all-in-one' delivery of *Geo*Cas9-gRNA (*Mir et al., 2018*). Until the very recent cryo-EM structures of *Geo*Cas9 (*Shen et al., 2024*; *Eggers et al., 2024*), little was known about specific residues that influence its structure, gRNA binding, or function. Our recent NMR work with *Spy*Cas9 uncovered pathways of micro-millisecond timescale motions that propagate chemical information related to allostery and specificity through *Spy*Rec and its RNA:DNA hybrid, (*Skeens et al., 2024*; *East et al., 2020*) prompting us to investigate this phenomenon in *Geo*Rec.

An atomic-level structural understanding of specificity in large multi-domain protein-nucleic acid complexes like Cas9 is often difficult to address by NMR spectroscopy. Although dynamic ensembles in DNA repair enzymes have provided some insight (*Lisi et al., 2017*), many efforts to improve Cas9 specificity and reduce off-target activity have relied on large mutational screens (*Slaymaker et al., 2016*) or error-prone PCR (*Vakulskas et al., 2018*), which are less intuitive. Inter-subunit allosteric

communication between the catalytic HNH domain and the Rec lobe is critical to Cas9 specificity, as the binding of off-target DNA sequences at Rec alter HNH dynamics to affect DNA cleavage (*Jinek et al., 2012*; *Guzman et al., 2015*; *Doudna and Charpentier, 2014*). To further probe the fundamental role of protein motions in the function and specificity of *Geo*Cas9, as well as the effect of protein-nucleic acid interactions on its structural signatures, we engineered two mutations in *Geo*Rec (K267E and R332A, housed within *Geo*Rec2). We hypothesized that these variants could enhance *Geo*Cas9 specificity (i.e. limit its off-target cleavage) for two reasons. First, the chosen mutation sites are homologous to those of specificity-enhancing variants of *Spy*Cas9 (*Vakulskas et al., 2018*; *Casini et al., 2018*). Second, altered Cas9-gRNA interactions have been shown to be a consequence of specificity-enhancement and these charged residues appear to directly interact with the gRNA (*Palermo et al., 2018*; *Dagdas et al., 2017*; *Chen et al., 2017*; *Ricci et al., 2019*). Balancing these two points is the fact that Type-II Cas systems generally have conserved nuclease domains but are delineated by highly varied Rec lobes (*Mir et al., 2018*). This implies that the structural and dynamic properties of Rec may play an outsized role in differentiating the functions of *Spy*Cas9 and *Geo*Cas9, which may not be identical. Nevertheless, our work provides new insight into the biophysical, biochemical, and functional role of the *Geo*Rec lobe and how mutations modulate the domain itself and its interaction with gRNA in full-length *Geo*Cas9.

## Results

### The structural similarity of GeoRec1, GeoRec2, and GeoRec facilitates NMR analysis of protein dynamics and RNA affinity

*Geo*Cas9 is a 1087 amino acid polypeptide, thus we employed a 'divide and concur' approach for NMR studies, which we previously showed to be useful for quantifying allosteric structure and motion in *Spy*Cas9 (*Skeens et al., 2024*; *East et al., 2020*; *Nierzwicki et al., 2021*; *Nierzwicki et al., 2022*). The *Geo*Rec lobe is comprised of subdomains *Geo*Rec1 and *Geo*Rec2, which likely work together to recognize nucleic acids. We engineered constructs of the *Geo*Rec1 (136 residues, 16 kDa) and *Geo*Rec2 (212 residues, 25 kDa) subdomains and solved the X-ray crystal structure of *Geo*Rec2 at 1.49 Å, which aligns remarkably well with the structure of the *Geo*Rec2 domain within the AlphaFold model (RMSD 1.03 Å) and new cryo-EM structure of *Geo*Cas9 (RMSD 1.10 Å, *Figure 1A*). We were neither able to crystallize *Geo*Rec1 nor full-length *Geo*Cas9 in the apo state, but our *Geo*Rec2 crystal structure represents the structure of the subdomain within the full-length *Geo*Cas9 protein quite well. Our previous studies of *Geo*HNH also show identical superpositions of X-ray crystal structures with full-length Cas complexes (*Belato et al., 2022a*). In addition to the individual subdomains, we also generated an NMR construct of the intact *Geo*Rec (370 residues, 43 kDa).

Despite only 22% sequence identity, the structure of *Spy*Rec3 and *Geo*Rec2 are highly similar (RMSD 2.00 Å, *Figure 1—figure supplement 1*). The structure of *Geo*Rec1, in contrast, does not align perfectly with *Spy*Rec1, instead, it partially aligns with both *Spy*Rec1 and *Spy*Rec2 (*Figure 1—figure supplement 1*). Thus, the nearly identical *Spy*Rec3 and *Geo*Rec2 architectures and their intrinsic dynamics may be a common thread among Type II Cas9s of different size and PAM preference. To capture atomic-level signatures of *Geo*Rec, we obtained well-resolved $^1$H-$^{15}$N NMR fingerprint spectra for all three protein constructs and assigned the amide backbones (*Figure 1—figure supplement 2*). $^1$H-$^{15}$N amide and $^1$H-$^{13}$CH$_3$ Ile, Leu, and Val (ILV)-methyl NMR spectra (*Figure 1B*, *Figure 1—figure supplement 3*) of *Geo*Rec overlay very well with those of its individual subdomains, suggesting that the linkage of subdomains within the full-length *Geo*Rec polypeptide does not alter their individual folds. Consistent with this observation, circular dichroism (CD) thermal unfolding profiles of *Geo*Rec1 ($T_m$ ~34 °C) and *Geo*Rec2 ($T_m$ = 61.50 °C) are distinct and occur as separate events in the unfolding profile of *Geo*Rec (*Figure 1—figure supplement 4*). The dumbbell shape of *Geo*Rec, with its two globular subdomains connected by a short flexible linker, is a likely contributor to these biophysical properties.

### Rationally designed GeoRec2 mutants do not substantially impact the GeoRec structure

To understand how the structure and gRNA interactions of *Geo*Cas9 can be modulated at the level of *Geo*Rec, we engineered two charge-altering point mutants in the *Geo*Rec2 subdomain, K267E

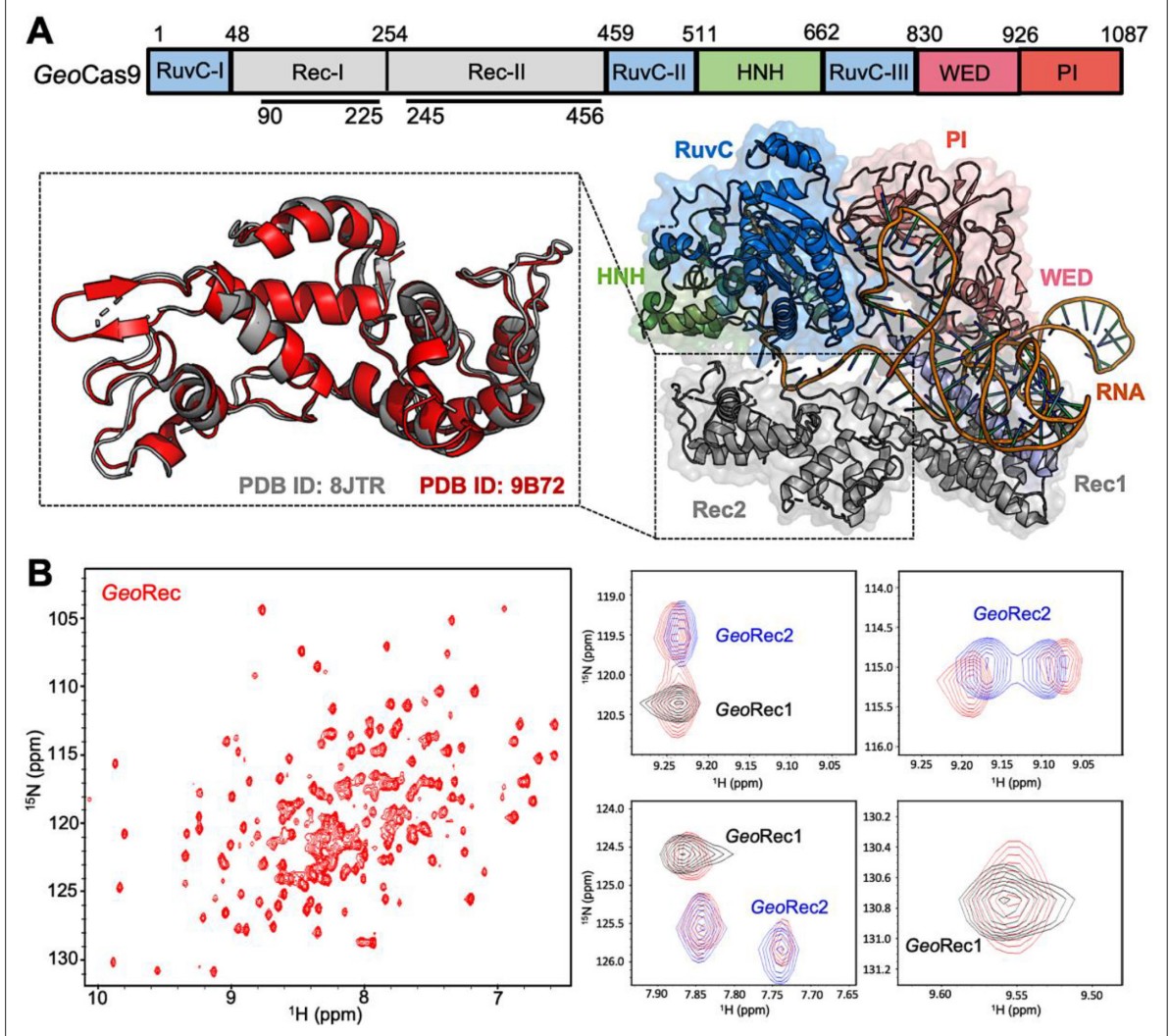

**Figure 1.** Architecture of *Geo*Cas9 and the *Geo*Rec domain. (**A**) Arrangement of *Geo*Cas9 domains across the primary sequence. The cryo-EM structure of *Geo*Cas9 in complex with gRNA (PDB: 8JTR) shows poor resolution of HNH. The *Geo*Rec2 domain from PDB: 8JTR (gray) is overlaid with our X-ray structure of *Geo*Rec2 (red, PDB: 9B72). (**B**) $^1$H$^{15}$N TROSY HSQC NMR spectrum of *Geo*Rec collected at 850 MHz. Overlays of this spectrum with resonances from spectra of *Geo*Rec1 (black) and *Geo*Rec2 (blue) demonstrate a structural similarity between the isolated subdomains and intact *Geo*Rec.

The online version of this article includes the following figure supplement(s) for figure 1:

**Figure supplement 1.** Sequence and structure analysis of *Geo*Rec and *Spy*Rec.

**Figure supplement 2.** Assigned $^1$H-$^{15}$N TROSY HSQC spectrum of *Geo*Rec1 (top left), *Geo*Rec (bottom left), and *Geo*Rec2 (bottom right).

**Figure supplement 3.** The dumbbell shape of *Geo*Rec is composed of the *Geo*Rec1and *Geo*Rec2 subdomains.

**Figure supplement 4.** Temperature-dependent CD unfolding profiles of *Geo*Rec1, *Geo*Rec2, and *Geo*Rec reveal that the unfolding profile of the individual subdomains are conserved within that of *Geo*Rec.

and R332A. Based on the AlphaFold2 model of *Geo*Cas9, both of these residues are <5 Å from the bound RNA:DNA hybrid and were predicted to interface with the nucleic acids directly (*Figure 2A/B*). A new experimental cryo-EM structure of *Geo*Cas9 confirmed the interaction between K267 and the gRNA, but does not report a<5 Å interaction of R332 with the gRNA (*Shen et al., 2024*). The rationale for our designed mutations was also that removal of positive charge would weaken the interactions between *Geo*Cas9 and the gRNA, affecting $K_d$ via the electrostatics or dynamics of the *Geo*Rec lobe. Studies of *Spy*Cas9 revealed that interaction of *Spy*Rec3 (analogous to *Geo*Rec2) with its RNA:DNA hybrid triggers conformational rearrangements that allow the catalytic HNH domain to sample its active conformation (*Dagdas et al., 2017*). Thus, *Spy*Rec3 acts as an allosteric effector that

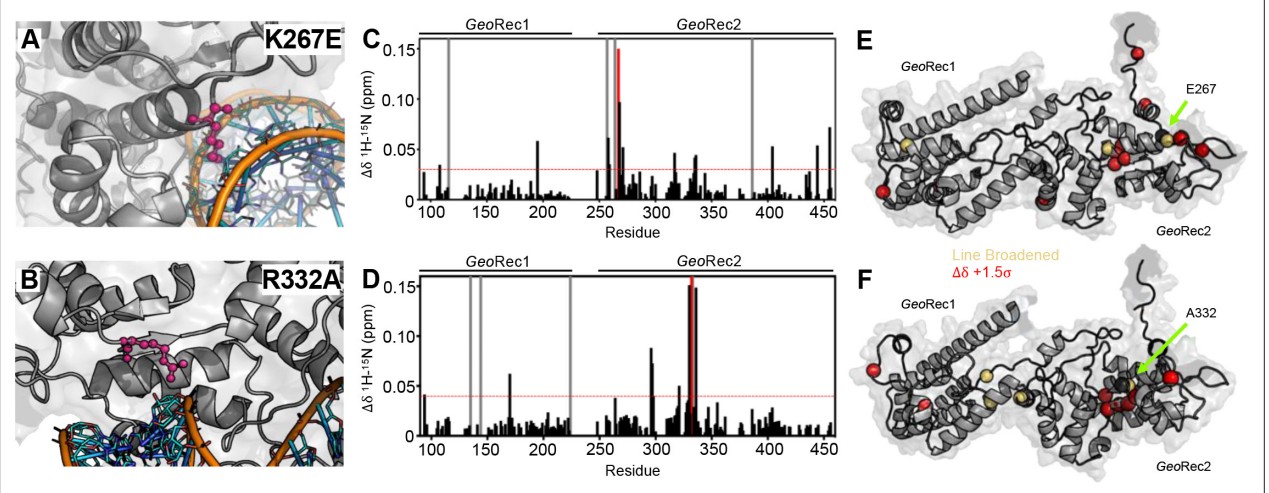

**Figure 2.** Impact of single-point mutations on the *Geo*Rec structure. (**A, B**) Sites of selected mutations within *Geo*Rec2, K267, and R332, are highlighted as purple sticks directly facing the RNA and DNA modeled from *Nme*Cas9 (PDB ID: 6JDV), allowing for prediction of the binding orientation within *Geo*Cas9. NMR chemical shift perturbations caused by the K267E (**C**) or R332A (**D**) mutations are plotted for each residue of *Geo*Rec. Gray bars denote sites of line broadening, the blue bar denotes an unassigned region of *Geo*Rec corresponding to the native Rec1-Rec2 linker, and the red bar indicates the mutation site. The red dashed line indicates 1.5σ above the 10% trimmed mean of the data. Chemical shift perturbations 1.5σ above the 10% trimmed mean are mapped onto K267E (**E**) and R332A (**F**) *Geo*Rec (red spheres). Resonances that have broadened beyond detection are mapped as yellow spheres and the mutation sites are indicated by a black sphere and green arrow.

The online version of this article includes the following figure supplement(s) for figure 2:

**Figure supplement 1.** Local impact of mutations on *Geo*Rec2.

**Figure supplement 2.** Secondary structure and stability of WT *Geo*Rec2 and variants.

recognizes the RNA:DNA hybrid to activate HNH. Mismatches (*i.e.* off-target DNA sequences) in the target DNA generally prevent *Spy*Rec3 from undergoing the full extent of its required conformational rearrangements, leaving HNH in a 'proofreading' state with its catalytic residues too far from the DNA cleavage site. Off-target DNA cleavage by Cas9 remains an area of intense study and substantial effort from various groups has gone into mitigating such effects (*Chen et al., 2017*; *Slaymaker et al., 2016*; *Ricci et al., 2019*; *Eggers et al., 2023*; *Doudna, 2020*). Indeed, many high-specificity *Spy*Cas9 variants contain mutations within *Spy*Rec3 that increase the threshold for its conformational activation, reducing the propensity for HNH to sample its active state in the presence of off-target DNA sequences (*Dagdas et al., 2017*; *Chen et al., 2017*; *Slaymaker et al., 2016*). Studies of flexibility within Rec itself, as well as its gRNA interactions in the presence of mutations, are therefore essential to connecting biophysical properties to function and specificity in related Cas9s.

The K267E *Geo*Rec2 variant is sequentially and structurally similar to a specificity enhancing site in *Spy*Cas9 (K526E), within the evoCas9 system (*Casini et al., 2018*). The *Spy*Cas9 K526E mutation substantially reduced off-target activity alone, but was even more effective in conjunction with three other single-point mutations in *Spy*Rec3 (*Casini et al., 2018*). The R332A *Geo*Rec2 variant also resembles one mutation within a high-specificity *Spy*Cas9 variant, an early iteration of HiFi *Spy*Cas9 called HiFi Cas9-R691A (*Vakulskas et al., 2018*). We assessed mutation-induced changes to local structure in *Geo*Rec via NMR chemical shift perturbations in $^1$H-$^{15}$N HSQC backbone amide spectra. Consistent with experiments using *Geo*Rec2 alone, chemical shift perturbations and line broadening are highly localized to the mutation sites. (*Figure 2C–F*). Perturbation profiles of the *Geo*Rec2 subdomain and intact *Geo*Rec also implicate the same residues as sensitive to the mutations (*Figure 2—figure supplement 1*).

CD spectroscopy revealed that wild-type (WT), K267E, and R332A *Geo*Rec2 maintained similar alpha-helical secondary structure, although the thermostability of both variants was slightly reduced from that of WT *Geo*Rec2 (*Figure 2—figure supplement 2*). The $T_m$ of WT *Geo*Rec2 is ~62 °C, consistent with the $T_m$ of the full-length *Geo*Cas9, while that of K267E *Geo*Rec2 was decreased to ~55 °C. Though the R332A *Geo*Rec2 $T_m$ remains ~62 °C, this variant underwent a smaller unfolding event near 40 °C before completely unfolding. These data suggest that despite small structural perturbations,

both mutations are destabilizing to *Geo*Rec2, which led us to expect a change in NMR-detectable protein dynamics.

## Mutations enhance and redistribute molecular motions within GeoRec2

Due to the high molecular weight of the intact *Geo*Rec lobe, decays in NMR signal associated with spin relaxation experiments were significant and hampered data quality. Thus, we focused on quantifying the molecular motions of the *Geo*Rec2 subdomain, where the K267E and R332A mutations reside, and the chemical shift perturbations are most apparent. To obtain high-quality per-residue information representative of *Geo*Rec, we measured longitudinal ($R_1$) and transverse ($R_2$) relaxation rates and heteronuclear $^1$H-[$^{15}$N] NOEs (*Figure 3—figure supplement 1*), then used these data in a Model-free analysis of per-residue order parameters ($S^2$). Previous measurements of $S^2$ across the adjacent *Geo*HNH nuclease revealed substantial ps-ns timescale flexibility, (*Belato et al., 2022a*) leading us to wonder whether a similar observation would be made for *Geo*Rec2, which abuts *Geo*HNH. Such a finding could suggest that HNH-Rec2 crosstalk in *Geo*Cas9 is driven primarily by rapid bond vector

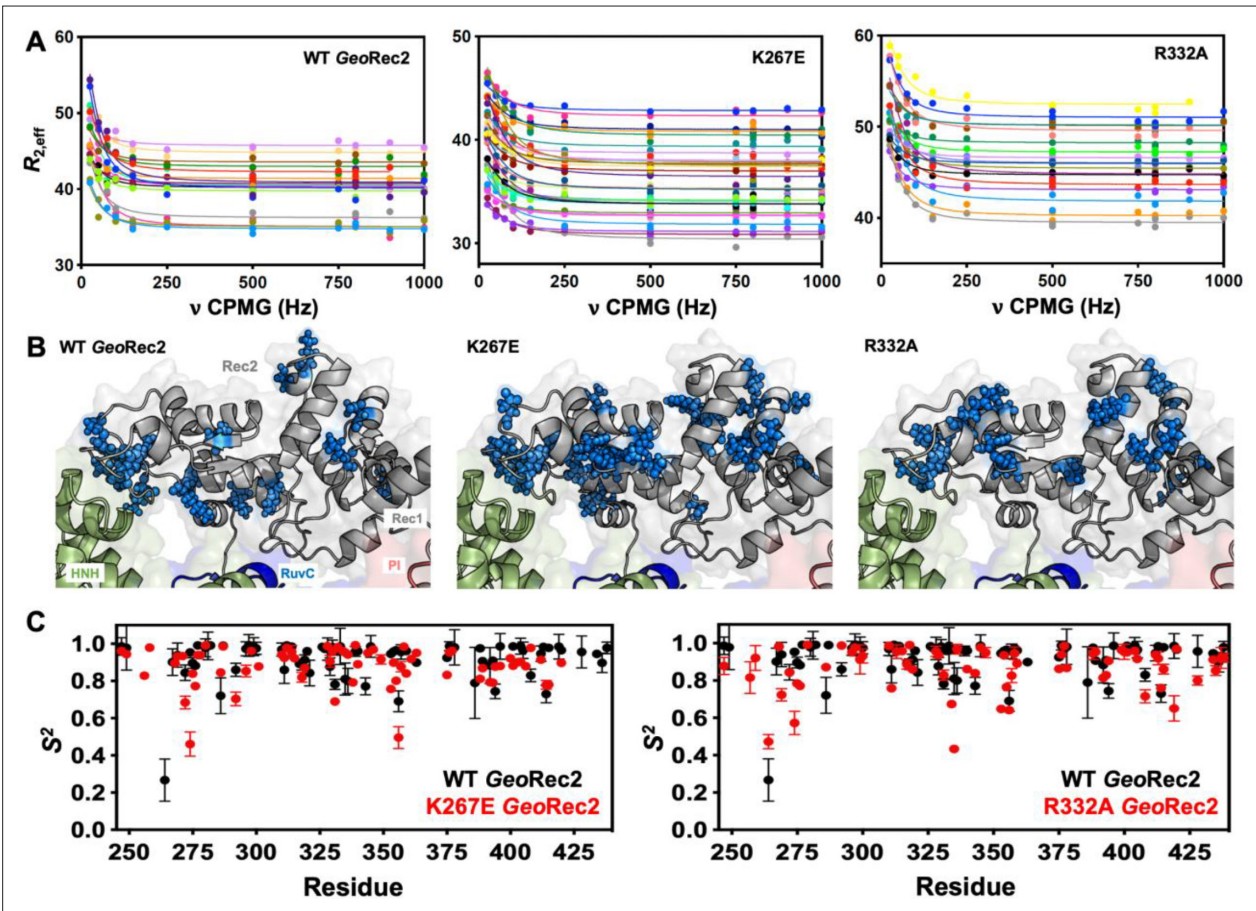

**Figure 3.** Single-point mutations enhance millisecond motions of *Geo*Rec2. (**A**) CPMG relaxation dispersion profiles of all residues with evidence of µs-ms motion, fit to a global $k_{ex}$ of 147±41 s$^{-1}$ (WT *Geo*Rec2, left), 376±89 s$^{-1}$ (K267E *Geo*Rec2, center), and 142±28 s$^{-1}$ (R332A *Geo*Rec2, right) collected at 25 °C and 600 MHz. Residues are colored in accordance with *Supplementary file 1*. Relaxation dispersion profiles for individual resonances are shown in *Figure 3—figure supplements 2–4*. (**B**) Sites exhibiting CPMG relaxation dispersion in (**A**) are mapped to *Geo*Rec as blue spheres. Adjacent domains within the cryo-EM structure of *Geo*Cas9 are also shown. (**C**) Per-residue NMR order parameters of WT (black), K267E, and R332A (red, separate plots) *Geo*Rec.

The online version of this article includes the following figure supplement(s) for figure 3:

**Figure supplement 1.** Effect of single-point mutations on fast timescale motions in *Geo*Rec2.

**Figure supplement 2.** CPMG relaxation dispersion curves collected at 25 °C and 600 MHz for WT *Geo*Rec2.

**Figure supplement 3.** CPMG relaxation dispersion curves collected at 25 °C and 600 MHz for K267E *Geo*Rec2.

**Figure supplement 4.** CPMG relaxation dispersion curves collected at 25 °C and 600 MHz for R332A *Geo*Rec2.

fluctuations. However, unlike *Geo*HNH, $S^2$ values for *Geo*Rec2 are globally elevated, suggesting that the ps-ns motions of this subdomain arise primarily from global tumbling of the protein in solution. We therefore carried out Carr-Purcell-Meiboom-Gill (CPMG) relaxation dispersion NMR experiments to assess the flexibility of *Geo*Rec2 on slower timescales, which has been linked to chemical information transfer in the well-studied *Spy*Cas9 (*Palermo et al., 2018*; *Skeens et al., 2024*; *East et al., 2020*; *Nierzwicki et al., 2021*). Evidence of µs-ms motions (i.e. curved relaxation dispersion profiles) is observed in 17 residues within the *Geo*Rec2 core, spanning its interfaces to Rec1 and HNH (*Figure 3A and B*, *Supplementary file 1*). Such motions are completely absent from *Geo*HNH, thus two neighboring domains, *Geo*Rec2 and *Geo*HNH, diverge in their intrinsic flexibility (at least in isolation), raising questions about the functional implications of these motions in *Geo*Rec2. We previously showed that heightened flexibility of *Spy*Rec3 via specificity-enhancing mutations concomitantly narrowed the conformational space sampled by *Spy*HNH, highlighting a 'motional trade-off' between the domains. Manipulation of the flexibility of *Spy*Cas9 and *Geo*Cas9 domains by mutagenesis also impacts aspects of nucleic acid binding and cleavage, (*Belato et al., 2022b*; *Palermo et al., 2018*; *Sternberg et al., 2015*; *Chen et al., 2017*; *Nierzwicki et al., 2021*) which led us to investigate similar perturbations in *Geo*Rec2.

Since *Spy*Rec3 and *Geo*Rec2 have similar structures and µs-ms flexibility, we speculated that charge-altering mutations would modulate the biophysical properties of *Geo*Rec and the function of *Geo*Cas9, as observed for *Spy*Cas9. We investigated K267E and R332A *Geo*Rec2 with NMR spin relaxation, as described for WT *Geo*Rec2 (*vide supra*). An analysis of chemical exchange rates, $k_{ex}$, derived from dual-field CPMG relaxation dispersion show a global $k_{ex}$ for WT *Geo*Rec2 of 147±41 s$^{-1}$. The K267E mutation, which directly contacts the nucleic acids, shifts the globally fitted $k_{ex}$ to 376±89 s$^{-1}$, while the R332A variant maintains a global $k_{ex}$ similar to that of WT *Geo*Rec2 (142±28 s$^{-1}$) and consistent with its similar thermal stability. The global fit of the K267E variant is based on CPMG profiles of 33 residues, while that of R332A is derived from 18 residues (*Figure 3—figure supplements 2–4*, *Supplementary file 1*). Interestingly, the residues participating in the global motions of both variants are distinct from those of WT *Geo*Rec2, demonstrating that residue-specific flexibility is redistributed throughout *Geo*Rec2, which suggests an altered intradomain molecular crosstalk within the larger *Geo*Rec. Indeed, perturbation to NMR-detectable motions in *Spy*Cas9 rewired its allosteric signaling and enzymatic function (*Skeens et al., 2024*; *Nierzwicki et al., 2021*). A similar dynamic modulation of *Geo*Cas9 may fine-tune its DNA cleavage, which has been demonstrated within the *Geo*HNH nuclease (*Belato et al., 2022b*) and wedge (WED) domains (*Eggers et al., 2023*). We also assessed the ps-ns fluctuations of *Geo*Rec2 variants (a negligible contribution to the WT *Geo*Rec2 dynamic profile) and calculated order parameters from $R_1$, $R_2$ and $^1$H-[$^{15}$N] NOE relaxation measurements (*Figure 3C*). Bond vector fluctuations on the ps-ns timescale are only locally altered, thus the mutation-induced reshuffling of these motions is negligible (<$\Delta S^2$>≤0.1) and suggests that, like WT *Geo*Rec2, ps-ns motion arises primarily from global tumbling in solution.

## Mutations within GeoRec alter its affinity for RNA

The role of the Rec lobe in orienting the RNA:DNA hybrid within Cas9 is crucial to its function (*Palermo et al., 2018*; *Dagdas et al., 2017*; *Mir et al., 2018*; *Jiang et al., 2015*). Thus, the structure, motions, and nucleic acid interactions of Rec represent a critical piece of the Cas9 signaling machinery. Previous studies of *Spy*Cas9 revealed that gRNA binding to the Rec lobe induces a global structural rearrangement of the protein that positions the adjacent HNH into its 'proofreading' state (*Chen et al., 2017*) after which target DNA binding positions the nucleases into active conformations for cleavage (*Chen et al., 2017*; *Palermo et al., 2017*). We wondered if the atomistic details of the apo *Geo*Cas9-to-RNP transition could be captured by NMR using the *Geo*Rec construct. In our previous studies, we used an in vitro DNA cleavage assay with *Geo*Cas9 and a 141nt gRNA containing a 21nt spacer targeting the mouse *Tnnt2* gene locus (*Belato et al., 2022b*). Since this assay was already established, we utilized the same gRNA sequence. However, truncating this gRNA was necessary to optimize binding studies for NMR analysis. We focused on the 5' end of the gRNA, which includes the spacer sequence, based on the *Geo*Cas9 AlphaFold2 model and structural data from *Nme*Cas9 and *Spy*Cas9 showing interactions between the Rec lobe and this region of the gRNA. The subsequent cryo-EM structure of *Geo*Cas9 corroborated this interaction (*Shen et al., 2024*). Initial attempts using a truncated 101nt portion of the full gRNA resulted in poor NMR spectra. An overlay of the $^1$H-$^{15}$N HSQC NMR spectra

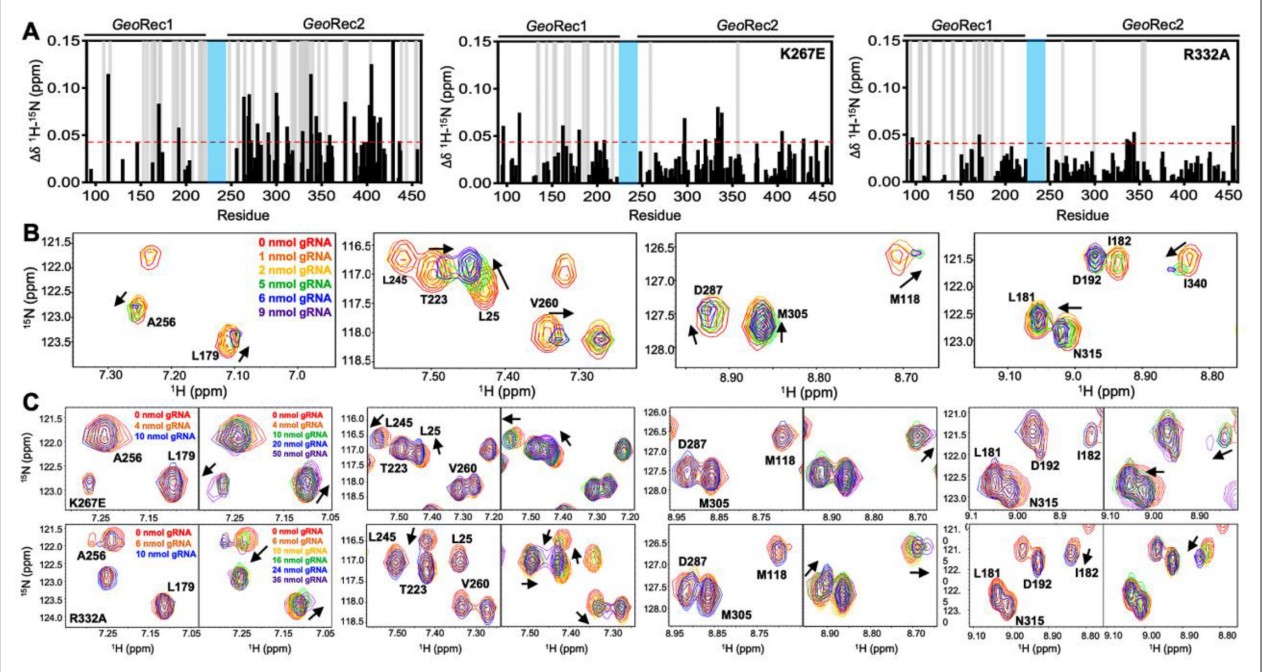

**Figure 4.** Mutations diminish the interaction between *Geo*Rec and RNA. (**A**) NMR chemical shift perturbations caused by RNA binding to WT, K267E, and R332A *Geo*Rec. Gray bars denote sites of line broadening, and the blue bar denotes an unassigned region of *Geo*Rec corresponding to the flexible Rec1-Rec2 linker. The red dashed line indicates 1.5σ above the 10% trimmed mean of the data. (**B**) Representative NMR resonance shifts caused by titration of a 39nt portion of the full gRNA into WT *Geo*Rec. (**C**) NMR titration of 39nt RNA into K267E (top) and R332A (bottom) *Geo*Rec. The left panel of each pair demonstrates that minimal change in NMR chemical shift or resonance intensity is apparent at RNA concentrations mimicking the WT titration. The right panel of each pair depicts the titration over a threefold wider concentration range of RNA, where shifts and line broadening are visible. Representative resonances are colored by increasing RNA concentration in the legend.

The online version of this article includes the following figure supplement(s) for figure 4:

**Figure supplement 1.** 850 MHz $^1$H$^{15}$N TROSY HSQC NMR spectra of apo-*Geo*Rec (red) and *Geo*Rec in complex with either a 101-nt.

**Figure supplement 2.** Structures of full-length.

**Figure supplement 3.** RNA-induced structural perturbations to WT and variant *Geo*Rec2.

**Figure supplement 4.** Analysis of RNA-induced chemical shift perturbations in WT *Geo*Rec2 and variants.

of apo *Geo*Rec and *Geo*Rec-RNP at a 1:1 molar ratio showed extensive line broadening (***Figure 4— figure supplement 1***), likely due to the large size of the complex (75.5 kDa). To mitigate this issue, a 39nt RNA containing the 21 bp spacer sequence was selected for its ability to maintain the NMR signal while being long enough to interact fully with the Rec lobe, as suggested by prior structures (***Figure 4—figure supplement 2***). When bound to *Geo*Rec, this complex is 55.6 kDa and a $^1$H-$^{15}$N NMR spectral overlay of apo *Geo*Rec and the domain bound to 39nt RNA shows clear, resolved resonances with significant chemical shift perturbations and line broadening (***Figure 4A/B***, ***Figure 4—figure supplement 1***). The strongest chemical shift perturbations are localized to the *Geo*Rec2 subdomain that interfaces with the RNA:DNA hybrid at the PAM distal end, where previous studies of specificity-enhancing variants of *Spy*Cas9 have identified alterations in nucleic acid binding to *Spy*Rec3 (***Skeens et al., 2024***). It is not known whether specific residues at the PAM distal binding interface of *Geo*Rec2 play a similar role. Line broadening is evident in both *Geo*Rec1 and *Geo*Rec2, primarily localized to the RNA:DNA hybrid interface revealed in recent *Geo*Cas9 structures. Microscale thermophoresis (MST) experiments quantified the affinity of *Geo*Rec for this RNA, producing a $K_d = 3.3 \pm 1.5$ µM that is consistent with the concentration-dependent NMR chemical shift perturbations (***Figure 5A***).

To understand how the K267E and R332A mutants impact RNA binding to *Geo*Rec, we conducted RNA titration experiments via NMR and observed that chemical shift perturbations were attenuated in both variants, relative to WT *Geo*Rec. Despite this muted structural effect, the impact from RNA-induced line broadening remains substantial in the *Geo*Rec1 subdomain. Our NMR data revealed that a three-fold greater concentration of RNA was required to induce the maximal structural and dynamic

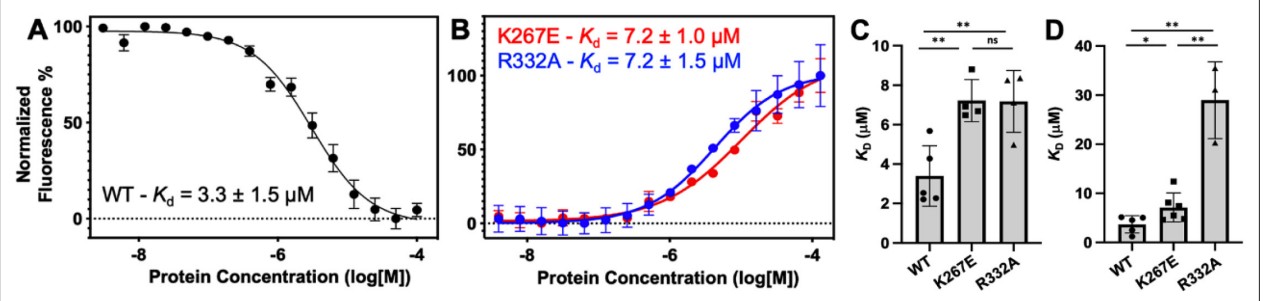

**Figure 5.** Representative MST-derived profiles of WT. (**A**), K267E, and R332A (**B**) *Geo*Rec binding to a Cy5-labeled 39nt portion of a gRNA, yielding $K_d$ = 3.3 ± 1.5 μM, $K_d$ = 7.2 ± 1.0 μM and $K_d$ = 7.2 ± 1.5 μM, respectively. Bar graphs comparing $K_d$ values across n≥3 technical replicate samples are shown for Tnnt2 RNA (**C**) and 8UZA RNA from a recent cryo-EM structure (**D**). *p<0.05, **p<0.004.

The online version of this article includes the following figure supplement(s) for figure 5:

**Figure supplement 1.** Stabilizing effect of gRNA on *Geo*Cas9.

effects in the variants than is required for WT *Geo*Rec (**Figure 4A/C**), suggesting that the variants have a reduced RNA affinity. MST experiments showed statistically significant reductions in RNA affinity for the K267E and R332A constructs, relative to WT *Geo*Rec, where K267E *Geo*Rec produced a $K_d$ = 7.2 ± 1.0 μM and R332A *Geo*Rec produced a $K_d$ = 7.2 ± 1.5 μM (**Figure 5B**, **Supplementary file 2**). The ~twofold increase in $K_d$ may also be due, in part, to a change in the binding mode of the RNA, such as a faster $k_{off}$. Collectively, these data reveal that mutations within *Geo*Rec primarily alter its structure around the mutation site with weaker distal effects, but more significantly impact protein dynamics and in turn, the RNA interaction. NMR experiments also demonstrate that the presence of RNA impacts both subdomains of *Geo*Rec, providing a significant structural interface for additional molecular tuning of nucleic acid binding.

To investigate the impact of RNA binding on *Geo*Rec2 in greater detail, we conducted NMR titration experiments using the isolated domain, which yielded even clearer NMR spectra. **Figure 4—figure supplement 3** shows NMR spectra of WT, K267E, and R332A *Geo*Rec2 overlaid with their corresponding RNA-bound spectra (39nt Tnnt2 RNA). At protein:RNA molar ratios used for full-length *Geo*Rec studies, the WT *Geo*Rec2 spectrum exhibited significant line broadening across the *Geo*Rec2 sequence. Plots of NMR peak intensities ($I_{bound}/I_{free}$) show substantial resonance intensity losses (**Figure 4—figure supplement 3**), with many residues likely in the intermediate exchange regime, in addition to the assumed changes in rotational correlation of the domain. In comparison, spectra of the RNA-bound K267E and R332A *Geo*Rec2 variants showed less pronounced signal decay at the same levels of titrant, retaining nearly double the $I_{bound}/I_{free}$ ratio across these spectra (**Figure 4—figure supplement 3**). These data are consistent with the results of NMR experiments with the 43 kDa *Geo*Rec, supporting the premise that *Geo*Rec2 mutations weaken its interaction with RNA. The less crowded NMR spectrum of isolated *Geo*Rec2 facilitated the resolution of distinct structural features that explain the impact of the mutations on RNA binding (**Figure 4—figure supplement 3**). For example, residue I53 adopts a similar conformation in RNA-bound WT and K267E *Geo*Rec2 but assumes a different structural state in R332A. Conversely, residue R25 populates a WT-like structure in RNA-bound R332A *Geo*Rec2, unlike K267E. Additionally, two resonances are observed for residue K71 in the RNA-bound R332A NMR spectrum, indicating real-time equilibration between two structural states. This effect is unique to the R332A variant and underscores subtle structural and dynamic changes to *Geo*Rec during RNA binding.

We further examined the NMR data to attempt to identify residues most critical for RNA binding to *Geo*Rec. In an overlay of the WT and mutant RNA-induced chemical shift perturbations (Δδ, **Figure 4—figure supplement 4**), it became clear that the effect of RNA binding to *Geo*Rec variants was muted, where even at saturating concentrations, the chemical shift perturbations across the K267E and R332A *Geo*Rec2 sequences were weaker than those of same residues in WT *Geo*Rec. The residual Δδ (WT - mutant) was plotted (**Figure 4—figure supplement 4**), where positive values indicate that residues in a *Geo*Rec variant are weakly affected by RNA, relative to WT. Negative residual Δδ denote sites where *Geo*Rec variants experience a greater structural impact from RNA than corresponding sites in WT. Of

particular interest are the positive residuals that hint at the sites in *Geo*Rec most critical for tight RNA binding. These residues were mapped onto the *Geo*Rec structure (*Figure 4—figure supplement 4*) and termed allosteric hotspots, as many are not at the RNA interface. Mutations of these hotspots in future studies offers a potential means of precisely tuning the affinity of *Geo*Rec to its gRNA. Notably, residues with positive residual Δδ (suggested as critical for tight RNA binding) largely overlap in the analysis of both variants. Specifically, residues F170, R192, H264, R269, L270, L279, H300, D301, E368, D376, D403, E405, E408, and I429 appear as allosteric hotspots (with CPMG relaxation dispersion) critical to WT-like RNA interaction.

Having observed a reduced affinity of *Geo*Rec variants for RNA by NMR and MST, we next quantified the impact of the K267E and R332A mutations on RNP formation and stability in full-length *Geo*Cas9. The thermal unfolding midpoint of full-length WT *Geo*Cas9 determined by CD is ~60 °C and the K267E and R332A mutations do not change the $T_m$ of the apo protein (*Figure 5—figure supplement 1*). Upon formation of an RNP (using a full-length gRNA), the $T_m$ of WT *Geo*Cas9 increases to 73 °C. K267E *Geo*Cas9 retains a similar $T_m$ increase to 70 °C, while R332A *Geo*Cas9 forms a less stable RNP with $T_m$ of 61 °C. The trend of these data is consistent with NMR and MST, which highlight that although K267E and R332A mutations within *Geo*Rec have somewhat muted structural effects, these changes alter protein dynamics and the interaction with gRNA.

## Mutations in full-length GeoCas9 alter its structural dynamics and interaction with gRNA

To further investigate the effects of mutations on protein dynamics, we performed molecular dynamics (MD) simulations based on the cryo-EM structure of full-length *Geo*Cas9 (PDB: 8UZA) in complex with gRNA and target DNA. We simulated the full-length WT *Geo*Cas9 and its K267E and R332A mutants as well as a double mutant combining K267E and R332A (*Figure 6A*), in three replicates of approximately 2 µs each. Multi-microsecond simulations revealed substantial changes in the dynamics of the *Geo*Cas9 mutants compared to the WT (*Figure 6—figure supplement 1*). Specifically, we observed that mutations in the REC domain significantly altered the dynamics of both the Rec and the adjacent HNH (*Figure 6—figure supplement 1*). Differential root-mean-square fluctuations (ΔRMSF) analysis of protein residues between the WT and variants further highlighted these alterations, showing increased dynamics in the HNH and Rec domains induced by the mutations (*Figure 6B*). To quantify the impact of these mutations, we analyzed protein-RNA interactions by calculating the number of contacts between *Geo*Cas9 and gRNA in the WT and mutant systems. A contact was defined as a distance between two atoms of ≤4.5 Å. The number of contacts was significantly reduced in all variants compared to the WT, with the most pronounced reduction observed in the K267E variant (*Figure 6C*). We next quantified gRNA-Rec domain binding by calculating the binding free energy using the MM-GBSA method over ~200 ns of stable simulation trajectories (details in Materials and methods). Consistent with a reduction in gRNA contacts, variants with the K267E mutation exhibited a substantial reduction in binding free energy (>60 kcal mol$^{-1}$) relative to the WT, whereas the R332A variant displayed a smaller reduction (<20 kcal mol$^{-1}$, *Figure 6D*). Notably, protein-DNA interactions remained largely unaffected, suggesting that these mutations do not impair *Geo*Cas9 DNA cleavage ability.

Additionally, we simulated a novel variant, i*Geo*Cas9 (PDB: 8UZB), containing mutations in the Rec1 and WED domains (*Figure 6A*, mutations highlighted in lime green). This variant was recently demonstrated to have enhanced specificity in genome-editing (*Eggers et al., 2024*). Intriguingly, i*Geo*Cas9 exhibited increased dynamics in the HNH and Rec domains, along with a reduction in gRNA binding free energy similar to the K267E and R332A mutants. These results suggest a distinct allosteric pathway involving additional residues that enables i*Geo*Cas9 to maintain improved DNA cleavage activity despite reduced gRNA binding affinity. In fact, i*Geo*Cas9 samples the greatest conformational space of any variant tested (*Figure 6—figure supplement 1*), suggesting a high level of flexibility is critical to enhanced specificity in *Geo*Cas9. Collectively, our simulations reveal that mutations K267E and R332A destabilize the *Geo*Cas9 interaction with gRNA, consistent with NMR observations. Furthermore, the enhanced dynamics and altered binding affinities observed in i*Geo*Cas9 indicate potential allosteric mechanisms that optimize its genome-editing functionality, where K267E and R332A evoke a similar, but lesser degree of biophysical change.

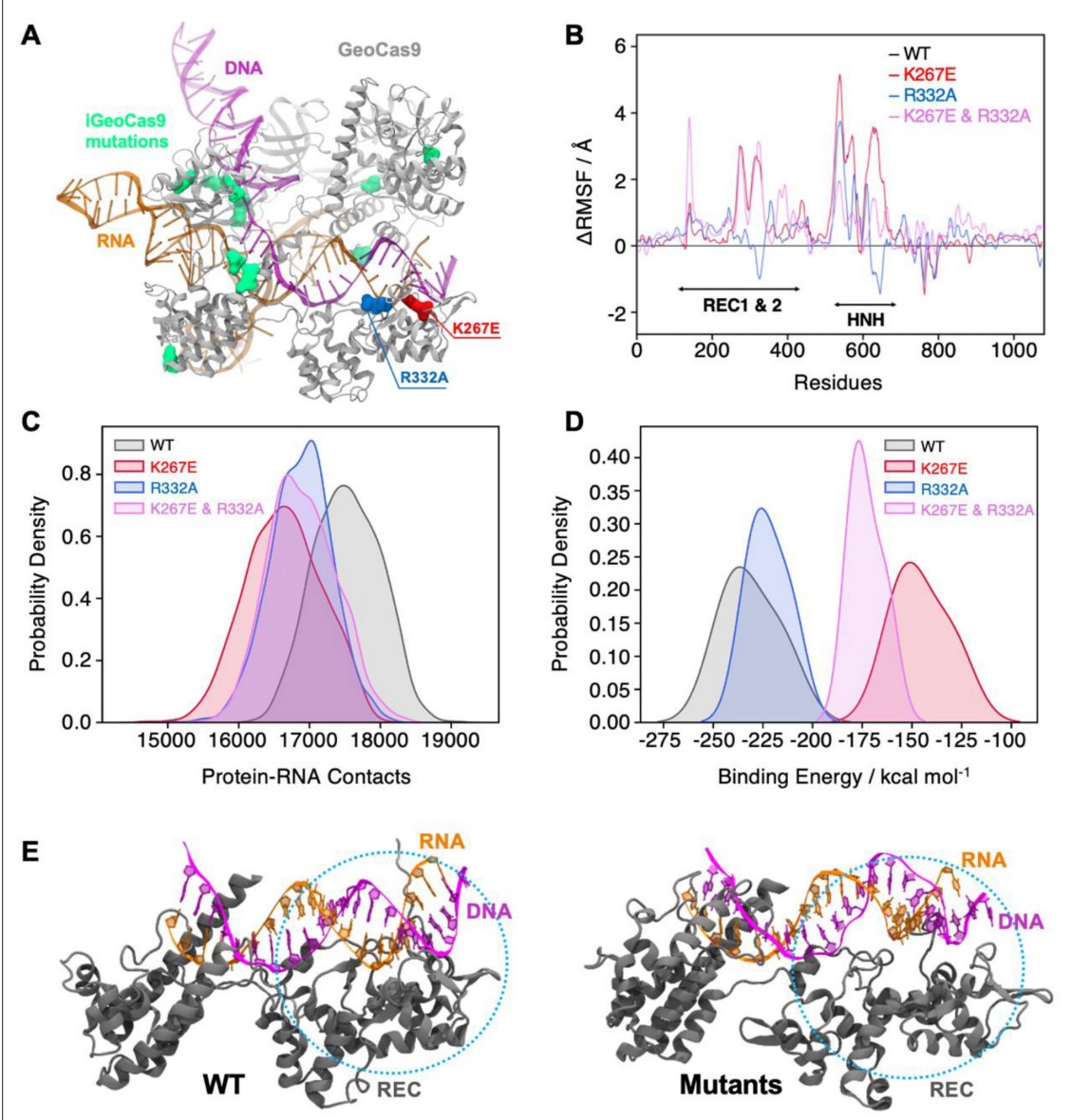

**Figure 6.** Effects of mutations in full-length *Geo*Cas9 revealed by MD simulations. (**A**) The structure of *Geo*Cas9 (PDB: 8UZA, protein in gray) bound to gRNA (orange) and DNA (magenta) is shown. Mutations studied include K267E (red), R332A (blue), and the 10 mutations of i*Geo*Cas9 (lime green), all highlighted in surface representation. (**B**) Differential root-mean-square fluctuations (ΔRMSF) of protein residues computed between WT *Geo*Cas9 and the K267E (red), R332A (blue), and double mutant (pink). (**C**) Distribution of protein-contacts for WT and *Geo*Cas9 variants computed over the 6 µs simulation ensemble. (**D**) Comparison of gRNA binding free energy to the Rec domain in WT *Geo*Cas9 and variants. (**E**) Representative snapshots from MD simulations illustrating structural changes in Rec-gRNA association in WT *Geo*Cas9 (left) and variants (right).

The online version of this article includes the following figure supplement(s) for figure 6:

**Figure supplement 1.** Analysis of MD simulations.

## DNA cleavage assays suggest the highly stable GeoCas9 is resistant to functional changes by K267E or R332A mutations

The dynamic impact of the *Geo*Rec mutations and their altered gRNA interactions at the biophysical level led us to speculate that either mutation incorporated into full-length *Geo*Cas9 would also alter its DNA cleavage function, especially at elevated temperatures where WT *Geo*Cas9 is most active. Temperature-dependent functional alterations were previously observed for single-point mutations within *Geo*HNH (*Belato et al., 2022b*). Although the K267E and R332A mutations slightly diminished on-target DNA cleavage by *Geo*Cas9, the effect was very subtle and these overall cleavage activities

followed the temperature dependence of WT *Geo*Cas9 quite closely (*Figure 7—figure supplement 1*).

To assess the impact of the K267E and R332A mutations on *Geo*Cas9 specificity, we assayed the propensity for off-target cleavage using DNA substrates with mismatches 5–6 or 19–20 base pairs from the PAM site (*Figure 7—figure supplement 2*, *Supplementary files 3 and 4*). As a control for on- and off-target activity, we assayed WT *Spy*Cas9 alongside the widely used high-specificity HiFi-*Spy*Cas9 variant (*Vakulskas et al., 2018*; *Figure 7—figure supplement 2*, *Supplementary file 4*) and found a lower percent of digested off-target (mismatched) DNA sequences when compared to WT *Spy*Cas9. As expected, WT *Geo*Cas9 was increasingly sensitive to mismatched target sequences closer to the seed site, which has been demonstrated with *Spy*Cas9 and other Cas systems (*Harrington et al., 2017*; *Lee et al., 2016*; *Jinek et al., 2012*; *Hou et al., 2013*). No significant differences in activity were observed with digestion durations ranging from 1 to 60 min (*Harrington et al., 2017*), implying that a 1-min digestion is sufficient for in vitro activity of *Geo*Cas9 with the target DNA template. While these findings generally align with prior investigations of off-target DNA cleavage (*Harrington et al., 2017*; *Lee et al., 2016*; *Jinek et al., 2012*), there are nuanced differences. Specifically, a previous study reported ~10% cleavage of off-target DNA with a mismatch 5–6 base pairs from the PAM by WT *Geo*Cas9 (*Harrington et al., 2017*). Our results showed nearly 50% cleavage for the same off-target mismatch, but still a significant decrease in cleavage from on-target or 19–20 base pair distal mismatches. This could be due to the relatively high RNP concentrations (600–900 nM) in our assay (for clear visibility on the gel), compared to prior studies with RNP concentrations ≤500 nM (*Harrington et al., 2017*). Our results corresponded closely to those of prior studies with a 19–20 base pair mismatch, where off-target cleavage is tolerated by WT *Geo*Cas9 (*Harrington et al., 2017*). Single-point mutants K267E and R332A *Geo*Cas9 have negligible impact on *Geo*Cas9 specificity (both variants follow the trend of WT *Geo*Cas9, *Figure 7—figure supplement 2*), which contrasts prior work with *Spy*Cas9 that demonstrated robust specificity enhancement with single-point mutations in Rec (*Vakulskas et al., 2018*; *Casini et al., 2018*). Additionally, a *Geo*Cas9 double mutant K267E/R332A exhibits decreased on-target cleavage efficiency, which has been noted in high-specificity Cas systems (*Chen et al., 2017*; *Slaymaker et al., 2016*; *Vakulskas et al., 2018*; *Casini et al., 2018*; *Kleinstiver et al., 2016*; *Hu et al., 2018*; *Lee et al., 2018*). However, the additive effect of the K267E/R332A double mutant still does not enhance *Geo*Cas9 specificity in our assay.

MST-derived binding affinities using full-length Tnnt2 gRNA and full-length WT, K267E, or R332A *Geo*Cas9 indicate that all three proteins have similar affinities for the gRNA used in the functional assays (*Figure 7*). Thus, mutations do not substantially alter full-length *Geo*Cas9 binding to Tnnt2 gRNA, supporting similar cleavage activities for these proteins. We repeated the MST experiments

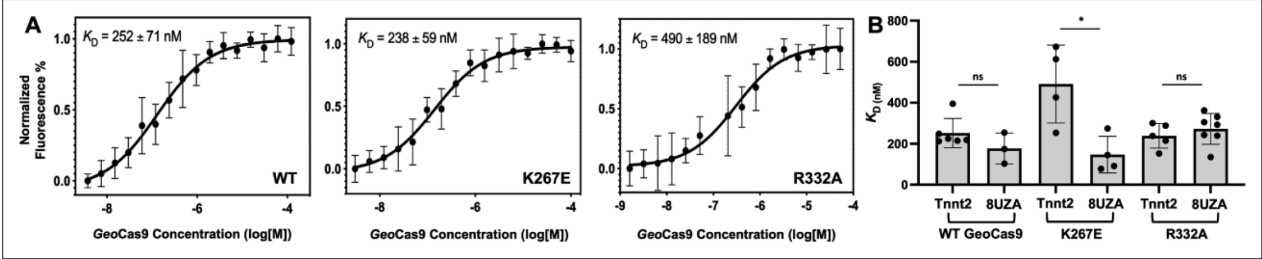

**Figure 7.** Binding affinities determined for two guide RNAs to *Geo*Cas9. (**A**) Representative MST-derived profiles of WT, K267E, and R332A *Geo*Cas9 binding to a Cy5-labeled full-length Tnnt2 gRNA. (**B**) Bar graph comparing $K_d$ values across n≥3 technical replicate samples are shown for Tnnt2 and 8UZA gRNA from a recent cryo-EM structure of *Geo*Cas9. *p<0.01.

The online version of this article includes the following source data and figure supplement(s) for figure 7:

**Figure supplement 1.** RNPs of varying concentrations of WT, K267E, or R332A *Geo*Cas9 and gRNA were incubated at 37, 60, 75, or 85 °C for 30 min, after which the RNPs were used for individual cleavage reactions.

**Figure supplement 1—source data 1.** Raw gel image of DNA cleavage by WT *Geo*Cas9, K267E *Geo*Cas9, and R332A *Geo*Cas9 at 37, 60, 75, and 85 °C.

**Figure supplement 1—source data 2.** Raw gel image of DNA cleavage by WT *Geo*Cas9, K267E *Geo*Cas9, and R332A *Geo*Cas9 at 37, 60, 75, and 85 °C, labelled.

**Figure supplement 2.** Assessment of off-target DNA cleavage by WT *Geo*Cas9 and variants in vitro.

using a different gRNA sequence derived from the new cryo-EM structure of *Geo*Cas9 (PDB:8UZA), which has a different spacer sequence. We observed a similar trend, as all *Geo*Cas9 variants exhibited comparable affinities for this gRNA. Our functional studies illustrate an apparent resilience of *Geo*Cas9 to major functional changes at the level of Rec, despite comparable mutations having profound functional impacts in mesophilic Cas9s.

## Discussion

CRISPR-Cas9 is a powerful tool for targeted genome editing with high efficiency and modular specificity (*Harrington et al., 2017*; *Chen et al., 2017*; *Vakulskas et al., 2018*; *Casini et al., 2018*). Allosteric signals propagate DNA binding information to the HNH and RuvC nuclease domains, facilitating their concerted cleavage of double-stranded DNA (*Jinek et al., 2012*; *Palermo et al., 2018*; *Sternberg et al., 2015*; *Chen et al., 2017*). The intrinsic flexibility of the nucleic acid recognition lobe plays a critical role in this information transfer, exerting a measure of conformational control over catalysis (*Palermo et al., 2018*). This study provides new insights into the structural, dynamic, and functional role of the thermophilic *Geo*Cas9 recognition lobe. Novel constructs of subdomains *Geo*Rec1 and *Geo*Rec2, as well as intact *Geo*Rec show a high structural similarity to the domains in full-length *Geo*Cas9, facilitating solution NMR experiments that captured the intrinsic allosteric motions across *Geo*Rec2. These studies revealed the existence of µs-ms timescale motions that are classically associated with allosteric signaling and enzyme function, which span the entire *Geo*Rec2 domain to its interfaces with *Geo*Rec1 and the adjacent *Geo*HNH domain.

Based on homology to specificity-enhancing variants of the better studied *Spy*Cas9, the biophysical and biochemical consequences of two mutations were tested in *Geo*Rec2, the larger *Geo*Rec lobe, and full-length *Geo*Cas9. We speculated that removing positively charged residues with potential to interact with negatively charged nucleic acids could disrupt *Geo*Cas9-gRNA complex formation, stability, and subsequent function by altering the protein or nucleic acid motions. Indeed, CPMG relaxation dispersion experiments revealed that mutations enhanced and reorganized the µs-ms flexibility of *Geo*Rec2. Further, NMR titrations showed the affinity of K267E and R332A *Geo*Rec for 39nt portions of two gRNAs to be weaker than that of WT *Geo*Rec, consistent with MST-derived $K_d$ values using the isolated domain. The mutations also diminished the stability of the full-length *Geo*Cas9 RNP complex, although this could result from an allosteric effect that destabilizes the *Geo*Cas9 structure without appreciably altering gRNA binding.

The collective changes to protein dynamics, gRNA binding, and RNP thermostability suggested that mutations could modulate *Geo*Cas9 function, as observed in similar studies of *Spy*Cas9 reporting that gRNA dynamics, affecting the potential for the RNA:DNA hybrid to dissociate, have affected function (*Dagdas et al., 2017*; *Chen et al., 2017*; *Ma et al., 2015*). Yet, the functional impact of single-point and double mutations in this work were negligible, despite homologous K-to-E and R-to-A single point mutations enhancing specificity of the mesophilic *Spy*Cas9. The biophysical impact of mutations within *Geo*Rec2 and *Geo*Rec may be tempered by its evolutionary resilience and the highly stable neighboring domains in the context of full-length *Geo*Cas9, reflected in an unchanged affinity for target DNA once the RNP was formed (*Nguyen et al., 2017*; *Saavedra et al., 2018*). Thus, a greater number of additive (or synergistic) mutations within *Geo*Rec would be required to fine-tune activity or specificity to a large degree.

It should be noted that the effects of these and other *Geo*Rec mutations may vary in vivo or with alternative target cleavage sites and cell types. Such studies will be the subject of future work, as will biochemical assays of homologous mutations across diverse Cas9s, which have contributed to the wide use of CRISPR technology (*Shmakov et al., 2017*). We also note that despite the homology between *Geo*Rec2 and *Spy*Rec3 and the latter's role in evo- and HiFi-*Spy*Cas9 variants that inspired the K267E and R332A mutations, the maximally enhanced *Spy*Cas9 variants contain four mutations each. Presumably each individual substitution plays a small role modulating specificity. However, there is no consistent pattern that discerns whether multiple mutations will have additive or synergistic impacts on Cas9 function. NMR and MD studies of high-specificity *Spy*Cas9 variants (HF-1, Hypa, and Evo, each with distinct mutations in the *Spy*Rec3 domain) reveal universal structural and dynamic variations in regions of *Spy*Rec3 that interface with the RNA;DNA hybrid (*Skeens et al., 2024*). Notably, a recently published variant, i*Geo*Cas9, *Eggers et al., 2023* demonstrated enhanced genome-editing capabilities in HEK293T cells with eight mutations, although none in the Rec2 subdomain. This study

highlighted the functional adaptability of i*Geo*Cas9 under low magnesium conditions, a trait beneficial in mammalian cells, distinguishing it from WT *Geo*Cas9. These very recently published data, as well as the findings reported here still advance our molecular understanding of the functional handles in *Geo*Cas9, relevant to the design of new enhanced variants.

This study marks the first phase of mapping allosteric motions and pathways of information flow in the *Geo*Rec lobe with solution NMR experiments. Such information transfer is critical to the crosstalk between Rec and HNH in several Cas9s. Despite NMR advancements in perdeuteration (*Venters et al., 1996*), transverse relaxation-optimized spectroscopy (TROSY; *Pervushin et al., 1997*) and sparse isotopic labeling (*Tugarinov et al., 2006*), per-residue dynamics underlying allosteric signaling in large multi-domain proteins such as *Geo*Cas9 (~126 kDa) have remained challenging to characterize. Novel cryo-EM structures of *Geo*Cas9 (*Eggers et al., 2023*) will facilitate the merging of future NMR and MD simulation studies to report on RNP dynamics and atomic level networks of communication. The identification of additional (or synergistic) allosteric hotspots within *Geo*Rec using an integrated workflow will help to further resolve the balance between structural flexibility and the unusually high stability of *Geo*Cas9, leading to new insight into targeted manipulation of RNA affinity and enhanced variants.

Here, we set out to biophysically characterize the Rec lobe of *Geo*Cas9 to obtain new understanding of its function (in the context of well-studied mesophilic Cas9s). Using an AlphaFold2 model, and later a cryo-EM structure of *Geo*Cas9, we introduced mutations based on proximal gRNA interactions and homology to specificity-enhancing sites in *Spy*Cas9. However, the mutations did not affect *Geo*Cas9 function as expected, highlighting the complicated interplay between the biophysics of mesophilic and thermophilic Cas enzymes and the difficulty of applying universal functional predictions to Cas9. The very recent report of the i*Geo*Cas9 variant further reinforces this point (*Eggers et al., 2024*). While high-specificity *Spy*Cas9 variants are heavily mutated in Rec3 (analogous to *Geo*Rec2), i*Geo*Cas9 lacks mutations in Rec2 entirely, raising new questions about the functional role of *Geo*Rec. MD simulations of WT *Geo*Cas9, i*Geo*Cas9, and the Rec variants revealed that while K267E and R332A induce dynamic effects on a similar trajectory to i*Geo*Cas9, a true high-specificity variant samples a very wide conformational space with displacements of both Rec and HNH (*Figure 6* and *Figure 6—figure supplement 1*). Through further study of the fundamental mechanism of *Geo*Cas9, it remains possible that engineering of *Geo*Rec may produce high-specificity variants.

## Materials and methods
### Expression and purification of *Geo*Rec1, *Geo*Rec2, *Geo*Rec, and *Geo*Cas9

The *Geo*Rec1 (residues 90–225) and *Geo*Rec2 (residues 245–456) subdomains, as well as the entire *Geo*Rec lobe (residues 90–456) of *G. stearothermophilus* Cas9 were engineered into a pET28a vector with a N-terminal His$_6$-tag and a TEV protease cleavage site. The K267E and R332A mutations were separately introduced into the *Geo*Rec2 plasmid. Plasmids were transformed into BL21 (DE3) cells (New England Biolabs). Protein samples for CD spectroscopy, MST, and functional assays were grown in Lysogeny Broth (LB, Fisher), while isotopically labeled samples for NMR were grown in M9 minimal media (deuterated for *Geo*Rec2 and *Geo*Rec) containing CaCl$_2$, MgSO$_4$, MEM vitamins, and 1.0 g/L $^{15}$N ammonium chloride and 2.0 g/L $^{13}$C glucose (Cambridge Isotope Laboratories), as the sole nitrogen and carbon sources, respectively. Cells were induced with 1 mM IPTG after reaching an OD$_{600}$ of 0.8–1.0 and grown for 4 hr at 37 °C post induction. The cells were harvested by centrifugation, resuspended in a buffer of 50 mM Tris-HCl, 250 mM NaCl, 5 mM imidazole, and 1 mM PMSF at pH 7.4, lysed by ultrasonication, and purified by Ni−NTA affinity chromatography. Following TEV proteolysis of the terminal His-tag, the samples were further purified on a Superdex75 size exclusion column. NMR samples were dialyzed into a buffer containing 20 mM NaPi, 80 mM KCl, 1 mM DTT, and 1 mM EDTA at pH 7.4.

The full-length *Geo*Cas9 plasmid was acquired from Addgene (#87700), expressed in TB media and was expressed and purified as previously described (*Harrington et al., 2017*). The K267E, R332A, and K267E/R332A variants were introduced into full-length *Geo*Cas9 by modifying the original plasmid acquired from Addgene.

## NMR spectroscopy

Backbone resonance assignments of *Geo*Rec1 and *Geo*Rec2 were carried out on a Bruker Avance NEO 600 MHz spectrometer at 25 °C. The following triple resonance experiments were collected for each sample: $^1$H-$^{15}$N TROSY-HSQC, HNCA, HN(CO)CA, HN(CA)CB, HN(COCA)CB, HN(CA)CO and HNCO. All spectra were processed in NMRPipe (*Delaglio et al., 1995*) and analyzed in Sparky (*Lee et al., 2015*). Three-dimensional correlations and assignments were made in CARA (*Keller, 2005*) and *Geo*Rec1 and *Geo*Rec2 backbone assignments were deposited in the BMRB under accession numbers 52363 and 51197, respectively. Backbone resonance assignments of *Geo*Rec were completed by transferring assignments from the individually assigned spectra of *Geo*Rec1 and *Geo*Rec2, as done previously for other large Cas9 fragments (*Nerli et al., 2021*; *De Paula et al., 2025*).

NMR spin relaxation experiments were carried out in a temperature-compensated manner at 600 and 850 MHz on Bruker Avance NEO and Avance III HD spectrometers, respectively. CPMG experiments were adapted from the report of Palmer and coworkers (*Loria et al., 1999*) with a constant relaxation period of 20ms and $\nu_{CPMG}$ values of 0, 25, 50, 75, 100, 150, 250, 500, 750, 800, 900, and 1000 Hz. Exchange parameters were obtained from global fits of the data carried out with RELAX (*Bieri et al., 2011*) using the R2eff, NoRex, and CR72 models, as well as in-house fitting in GraphPad Prism with the following models:

Model 1: No exchange

$$R_2^{eff} = R_2^0 \tag{1}$$

Model 2: Two-state, fast exchange (Meiboom equation *Luz and Meiboom, 1963*)

$$R_2^{eff} = R_2^0 + \frac{\phi}{k_{ex}} \left[ 1 - \frac{4_{CPMG}}{k_{ex}} \tan h \left( \frac{k_{ex}}{4_{CPMG}} \right) \right] \tag{2}$$

Global fitting of CPMG profiles was determined to be superior to individual fits based on the Akaike Information Criterion (*Cavanaugh and Neath, 2019*). Uncertainties in these rates were determined from replicate spectra with duplicate relaxation delays of 0, 25, 50 (×2), 75, 100, 150, 250, 500 (×2), 750, 800 (×2), 900, and 1000 Hz.

Longitudinal and transverse relaxation rates were measured with randomized $T_1$ delays of 0, 20, 60, 100, 200, 600, 800, and 1200ms and $T_2$ delays of 0, 16.9, 33.9, 50.9, 67.8, 84.8, and 101.8ms. Peak intensities were quantified in Sparky and the resulting decay profiles were analyzed in Sparky with errors determined from the fitted parameters. Uncertainties in these rates were determined from replicate spectra with duplicate relaxation delays of 20 (x2), 60 (x2), 100, 200, 600 (x2), 800, and 1200ms for $T_1$ and 16.9, 33.9 (x2), 50.9 (x2), 67.8 (x2), 84.8, 101.8 (x2) ms for $T_2$. Steady-state $^1$H-[$^{15}$N] NOE were measured with a 6 s relaxation delay followed by a 3 s saturation (delay) for the saturated (unsaturated) experiments and calculated by $I_{sat}/I_{ref}$. All relaxation experiments were carried out in a temperature-compensated interleaved manner.

Model-free analysis was carried out by fitting relaxation rates to five different forms of the spectral density function with local $\tau_m$, spherical, prolate spheroid, oblate spheroid, or ellipsoid diffusion tensors (*Brüschweiler et al., 1995*; *Mandel et al., 1995*; *Fushman et al., 1997*; *Orekhov et al., 1999*; *Korzhnev et al., 2001*; *Zhuravleva et al., 2004*). The criteria for inclusion of resonances in the diffusion tensor estimate was based on the method of Bax and coworkers (*Tjandra et al., 1995*). N-H bond lengths were assumed to be 1.02 Å and the $^{15}$N chemical shift anisotropy tensor was –160 ppm. Diffusion tensor parameters were optimized simultaneously in RELAX under the full automated protocol (*Bieri et al., 2011*). Model selection was iterated until tensor and order parameters did not deviate from the prior iteration.

NMR titrations were performed on a Bruker Avance NEO 600 MHz spectrometer at 25 °C by collecting a series of $^1$H-$^{15}$N TROSY HSQC spectra with increasing ligand (*i.e.* RNA) concentration. The $^1$H and $^{15}$N carrier frequencies were set to the water resonance and 120 ppm, respectively. Samples of WT, K267E, and R332A *Geo*Rec were titrated with a 39nt portion of a gRNA until no further spectral perturbations were detected. NMR chemical shift perturbations were calculated as:

$$\Delta \delta = \sqrt{\left( \Delta \delta_{HN}^2 + \Delta \delta_{NH}^2 / 25 \right) / 2}$$

## Microscale thermophoresis (MST)

MST experiments were performed on a Monolith X instrument (NanoTemper Technologies), quantifying WT, K267E, R332A, and K267E/R332A *Geo*Rec binding to a 39-nt Cy5-labeled RNA at a concentration of 20 nM in a buffer containing 20 mM sodium phosphate, 150 mM KCl, 5 mM MgCl$_2$, and 0.1% Triton X-100 at pH 7.6. The *Geo*Rec proteins were serially diluted from a 200 µM stock into 16 microcentrifuge tubes and combined in a 1:1 molar ratio with serially diluted gRNA from a 40 nM stock. After incubation for 5 min at 37 °C in the dark, each sample was loaded into a capillary for measurement. $K_d$ values for the various complexes were calculated using the MO Control software (NanoTemper Technologies). Statistical significance was calculated using a two-tailed T-test.

## Circular dichroism (CD) spectroscopy

All *Geo*Cas9 and *Geo*Rec proteins were buffer exchanged into a 20 mM sodium phosphate buffer at pH 7.5, diluted to 1 µM, and loaded into a 2 mm quartz cuvette (JASCO instruments). A CD spectrum was first measured between 200 and 250 nm, after which the sample was progressively heated from 20–90 °C in 1.0 °C increments while ellipticity was monitored at 222 and 208 nm. Phosphate buffer baseline spectra were subtracted from the sample measurements. Prior to CD measurements, *Geo*Cas9-RNP was formed by incubating 3 µM *Geo*Cas9 with its gRNA at a 1:1.5 molar ratio at 37 °C for 10 min. The unfolding CD data was fit in GraphPad Prism to:

$$Ellipticity\ (T) = \frac{\left[(m_f T + b_f) + (m_u T + b_u)\right] exp\left[\left(-\frac{\Delta H_{D,vH}}{R}\right)\left(\frac{1}{T} - \frac{1}{T_m}\right)\right]}{1 + exp\left[\left(-\frac{\Delta H_{D,vH}}{R}\right)\left(\frac{1}{T} - \frac{1}{T_m}\right)\right]}$$

## X-ray crystallography

*Geo*Rec protein purified as described above was crystallized by sitting drop vapor diffusion at room temperature by mixing 1.0 µL of 15 mg/mL *Geo*Rec in a buffer of 20 mM HEPES and 100 mM KCl at pH 7.5 with 2.0 µL of crystallizing condition: 0.15 M calcium chloride, 15% polyethylene glycol 6000, 0.1 M HEPES at pH 7.0. Crystals were cryoprotected in crystallizing condition supplemented with 30% ethylene glycol. Diffraction images were collected at the NSLS-II AMX beamline at Brookhaven National Laboratory under cryogenic conditions. Images were processed using XDS (*Kabsch, 2010*) and Aimless in CCP4 (*Winn et al., 2011*). Chain A of the *N. meningitidis* Cas9 X-ray structure (residues 249–445 only, PDB ID: 6JDQ) was used for molecular replacement with Phaser followed by AutoBuild in Phenix (*Liebschner et al., 2019*). Electron density was only observed for the *Geo*Rec2 subdomain. The *Geo*Rec2 structure was finalized through manual building in Coot (*Emsley et al., 2010*) and refinement in Phenix.

## Molecular dynamics (MD) simulations

Molecular Dynamics (MD) simulations were based on the cryo-EM structure of full-length *Geo*Cas9 (PDB: 8UZA, resolution 3.17 Å) in complex with gRNA and target DNA with two mutations in *Geo*Cas9 (at residues 8 and 582). Four systems were considered for the MD studies: WT, K267E, R332A, K267E/R332A and i*Geo*Cas9. We generated the WT *Geo*Cas9 by back-mutating A8D and A582H from the cryo-EM structure (PDB: 8UZA), followed by introducing the mutations K267E, R332A, or a double mutation (with both K267E and R332A) for the variant systems. Subsequently, we performed MD simulation of i*Geo*Cas9 (PDB: 8UZB, resolution 2.63 Å) consisting of 10 mutations (D8A, E149G, T182I, N206D, P466Q, H582A, Q817R, E843K, E854G, K908R). All systems were solvated with explicit water in a periodic box of ~134 Å x~154 Å x~151 Å resulting in ~276,000 atoms. Counter ions were added to neutralize the systems. MD simulations were performed using a protocol tailored for protein-nucleic acid complexes (*Sinha et al., 2023*), previously applied in studies of CRISPR-Cas systems (*Saha et al., 2024*; *Arantes et al., 2024*; *Sinha et al., 2024*). All the simulations were performed by using Amber ff19SB force field for protein (*Tian et al., 2020*), ff99bsc1 corrections and $\chi$ OL3 corrections for DNA and RNA, respectively (*Galindo-Murillo et al., 2016*; *Zgarbová et al., 2011*). Water molecules were described by TIP3P model (*Jorgensen et al., 1983*). All bonds involving hydrogens were constrained using the LINCS algorithm. A particle mesh Ewald method (PME) with a 10 Å cutoff was used to calculate electrostatics. Energy minimization was performed to relax the water molecules and counterions,

keeping the protein-nucleic acid complex fixed with harmonic potential restraints of 100 kcal/mol Å². Equilibration was performed by gradually increasing the temperature from 0 to 100 K and then to 200 K in canonical NVT ensemble and isothermal-isobaric NPT ensemble. A final temperature of 300 K was maintained via Langevin dynamics with a collision frequency $\gamma$=1/ps and a reference pressure of 1 atm was achieved through Berendsen barostat. Production runs were carried out in NVT ensemble for 2 µs for each system in three replicates, resulting in 6 µs per system (totaling 30 µs for all systems). The equations of motion were integrated with the leapfrog Verlet algorithm with a time step of 2 fs. All simulations were conducted using the GPU-empowered version of AMBER 22 (*Case et al., 2005*). Analysis was performed on the aggregate ensemble (i.e. ~6 µs per system).

To characterize the protein-nucleic acid interactions in all the systems under investigation, we performed contact analysis. A contact was considered to form between two atoms within a cutoff distance of ≤4.5 Å. The binding free energy of gRNA and DNA with *Geo*Cas9 was calculated using the Molecular Mechanics Generalized Born Surface Area (MM-GBSA) method (*Mongan et al., 2007*; *Nguyen et al., 2015*; *Nguyen et al., 2013*) This approach was used to compare the Rec-gRNA binding affinity of WT *Geo*Cas9 with its mutants. For each system, the binding energies were calculated over the ~200 ns ensemble of the stable trajectories at an interval of ~20 ns.

## DNA cleavage assays

*Geo*Cas9 gRNA templates containing 21-nt spacers targeting the mouse *Tnnt2* gene locus were introduced into EcoRI and BamHI sites in pUC57 (Genscript). The plasmid was transformed into BL21(DE3) cells (New England BioLabs) and subsequent restriction digest of the plasmid DNA was carried out using the BamHI restriction enzyme (New England BioLabs) according to the manufacturer's instructions. Linearized plasmid DNA was immediately purified using the DNA Clean and Concentrator-5 kit (Zymo Research) according to the manufacturer's instructions. RNA transcription was performed in vitro with the HiScribe T7 High Yield RNA Synthesis Kit (New England BioLabs). DNA substrates containing the target cleavage site (479 base pairs) were produced by polymerase chain reaction (PCR) using mouse genomic DNA as a template and primer pairs 5'-CAAAGAGCTCCTCGTCCAGT-3' and 5'-ATGGACTCCAGGACCCAAGA-3' followed by a column purification using the NucleoSpin® Gel and PCR Clean-up Kit (Macherey-Nagel). For the in vitro activity assay, RNP formation was achieved by incubating 3 µM *Geo*Cas9 (WT, K267E, R332A, or K267E/R332A mutant) and 3 µM gRNA at either 37 °C, 60 °C, 75 °C, or 85 °C for 30 min in a reaction buffer of 20 mM Tris, 100 mM KCl, 5 mM MgCl₂, 1 mM DTT, and 5% glycerol at pH 7.5. The 10 µL cleavage reactions were set up by mixing RNP at varying concentrations with 149 nanograms of PCR products on ice followed by incubation at 37 °C for 30 min. The reaction was quenched with 1 µL of proteinase K (20 mg/mL) and subsequent incubation at 56 °C for 10 min. 6 x DNA loading buffer was added to each reaction and 10 µL of reaction mixture per lane was loaded onto an agarose gel. DNA band intensity measurements were carried out with ImageJ.

For in vitro off-target activity assays, RNP formation was achieved by incubating 10 µM *Geo*Cas9 (WT, K267E, R332A, or K267E/R332A mutant) and 10 µM gRNA at 37 °C for 30 min in the reaction buffer described above. The 10 µL cleavage reactions were set up by mixing 1 µM RNP with 150 nanograms of PCR products (off-target DNA sequences listed in *Supplementary file 3*) on ice followed by incubation at 37 °C for varying time points. The reaction was quenched with 1 µL of proteinase K (20 mg/mL) and subsequent incubation at 56 °C for 10 min. 6 x DNA loading buffer was added to each reaction and 10 µL of reaction mixture per lane was loaded onto an agarose gel. DNA band intensity measurements were carried out with ImageJ. WT and HiFi *Spy*Cas9 control proteins were purchased from Integrated DNA Technologies (IDT, cat. No. 108158 and No. 108160, respectively), as was the associated *Spy*Cas9 gRNA, Alt-R CRISPR-Cas9 gRNA, with an RNA spacer sequence complementing 5'-TGGACAGAGCCTTCTTCTTC-3'. The on-target and off-target DNA sequences used for the *Spy*Cas9 in vitro cleavage assay can be found in *Supplementary file 4*.

## Acknowledgements

This work was supported by NIH grant R01 GM 136815 (to GP and GPL) and NSF grant MCB 2143760 (to GPL). GP acknowledges support from the NIH (Grant No. R01GM141329) and the NSF (CHE-2144823), as well as from the Sloan Foundation (FG-2023–20431) and the Camille and Henry Dreyfus Foundation (TC-24–063). This research used the AMX beamline of the National Synchrotron Light

Source II, a U.S. Department of Energy (DOE) Office of Science User Facility operated for the DOE Office of Science by Brookhaven National Laboratory under Contract No. DE-SC0012704. The Center for BioMolecular Structure (CBMS) is primarily supported by NIGMS through a Center Core P30 Grant (P30 GM133893), and by the DOE Office of Biological and Environmental Research (KP1607011). Computational studies were carried out using Expanse at the San Diego Supercomputing Center through allocation MCB160059 and Bridges2 at the Pittsburgh Supercomputer Center through allocation BIO230007 from the Advanced Cyberinfrastructure Coordination Ecosystem: Services & Support (ACCESS) program, which is supported by NSF grants #2138259, #2138286, #2138307, #2137603, and #2138296.

# Additional information

## Funding

| Funder | Grant reference number | Author |
|---|---|---|
| National Institutes of Health | R01GM136815 | George P Lisi |
| National Science Foundation | MCB2143760 | George P Lisi |
| National Institutes of Health | R01GM141329 | Giulia Palermo |
| National Science Foundation | CHE2144823 | Giulia Palermo |
| Alfred P. Sloan Foundation | FG-2023-20431 | Giulia Palermo |
| Camille and Henry Dreyfus Foundation | TC-24-063 | Giulia Palermo |

The funders had no role in study design, data collection and interpretation, or the decision to submit the work for publication.

## Author contributions

Helen B Belato, Alexa L Knight, Formal analysis, Investigation, Writing – original draft, Writing – review and editing; Alexandra M D'Ordine, Chinmai Pindi, Zhiqiang Fan, Formal analysis, Investigation, Writing – original draft; Jinping Luo, Formal analysis, Supervision, Investigation, Writing – original draft; Giulia Palermo, Conceptualization, Data curation, Funding acquisition, Writing – original draft, Writing – review and editing; Gerwald Jogl, Data curation, Supervision, Project administration, Writing – review and editing; George P Lisi, Conceptualization, Data curation, Supervision, Writing – original draft, Project administration, Writing – review and editing

## Author ORCIDs

George P Lisi https://orcid.org/0000-0001-8878-5655

Reviewer #1 (Public review): https://doi.org/10.7554/eLife.99275.4.sa1
Reviewer #2 (Public review): https://doi.org/10.7554/eLife.99275.4.sa2
Reviewer #3 (Public review): https://doi.org/10.7554/eLife.99275.4.sa3
Author response https://doi.org/10.7554/eLife.99275.4.sa4

# Additional files

## Supplementary files

Supplementary file 1. Residues fit to a global $k_{ex}$ in $^1$H-$^{15}$N CPMG relaxation dispersion analysis of WT, K267E, and R332A *GeoRec2*. A very small number of other resonances displaying curved CPMG profiles could not be globally fit and were excluded from this list. In all samples (*i.e.* WT, K267E, and R332A), the global fit was found to be the best statistical model.

Supplementary file 2. Guide RNA sequences used in *Geo*Cas9 MST measurements. The 39-nucleotide sequence of RNA used in MST and NMR studies of isolated *Geo*Rec is underlined.

Supplementary file 3. Nucleic acid sequences used in the *Geo*Cas9 in vitro off-target assay. The 23 base pair spacer sequence of gRNA is underlined. The spacer sequence within the DNA sequences is highlighted yellow, and the PAM are highlighted in blue.

Supplementary file 4. DNA sequences used in the *Sp*Cas9 in vitro off-target assay. Sites of mismatched DNA are highlighted in red.

MDAR checklist

## Data availability

NMR resonance assignments have been deposited in the BMRB under accession codes 51197 and 52363. All other data generated during this study are included in the manuscript and supporting files.

The following datasets were generated:

| Author(s) | Year | Dataset title | Dataset URL | Database and Identifier |
|---|---|---|---|---|
| Belato H, Knight A, D'Ordine A, Fan Z, Luo J, vG Jogl, Lisi GP | 2024 | 1H, 15N Backbone Assignments of Rec3 from GeoCas9 | https://bmrb.io/data_library/summary/index.php?bmrbId=51197 | Biological Magnetic Resonance Data Bank, BMR51197 |
| Knight A, D'Ordine A, Fan Z, Luo J, Jogl G, Lisi GP | 2024 | GeoRec1 backbone assignments | https://bmrb.io/data_library/summary/index.php?bmrbId=52363 | Biological Magnetic Resonance Data Bank, BMR52363 |

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
