## [Editor Report · eLife Assessment]

This study offers **valuable** insights into the conformational dynamics of the nucleic acid recognition lobe of GeoCas9, a thermophilic Cas9 from Geobacillus stearothermophilus. The authors investigate the influence of local dynamics and allosteric regulation on guide RNA binding affinity and DNA cleavage specificity through molecular dynamics simulations, advanced NMR techniques, RNA binding studies, and mutagenesis. While the mutations studied do not lead to significant changes in GeoCas9 cleavage activity, the study provides **convincing** evidence for the role of allosteric mechanisms and interdomain communication in Cas9 enzymes, and will be of great interest to biochemists and biophysicists exploring these complex systems.

---

## [Referee Report · Reviewer #1 (Public review)]

Summary:

In this study from Belato, Knight and co-workers, the authors investigated the Rec domain of a thermophilic Cas9 from Geobacillus stearothermophilus (GeoCas9). The authors investigated three constructs, two individual subdomains of Rec (Rec1 and Rec2) and the full Rec domain. This domain is involved in binding to the guide RNA of Cas9, as well as the RNA-DNA duplex that is formed upon target binding. The authors performed RNA binding and relaxation experiments using NMR for the wild-type domain as well as two-point mutants. They observed differences in RNA binding activities as well as the flexibility of the domain. The authors also performed molecular dynamics and functional experiments on full-length GeoCas9 to determine whether these biophysical differences affect the RNA binding or cleavage activity. Although the authors observed some changes in the thermal stability of the mutant GeoCas9-gRNA complex, they did not observe substantial differences in the guide RNA binding or cleavage activities of the mutant GeoCas9 variants.

Overall, this manuscript provides a detailed biophysical analysis of the GeoCas9 Rec domain. The NMR assignments for this construct should prove very useful, and can serve as the basis for future similar studies of GeoCas9 Rec domain mutants. While the two mutants tested in the study did not produce significant differences from wild-type GeoCas9, the study rules out the possibility that analogous mutations can be translated between type II-A and II-C Cas9 orthologs. Together, these findings may provide the grounds for future engineering of higher fidelity variants of GeoCas9

---

## [Referee Report · Reviewer #2 (Public review)]

The manuscript from Belato et al., used advanced NMR approaches and a mutagenesis campaign probe the conformational dynamics of the recognition lobe (Rec) of the CRISPR Cas9 enzyme from G. stearothermophilus (GeoCas9). Using truncated and full-length constructs they assess the impacts of two different point mutations have on the redistribution and timescale of these motions and assess gRNA recognition and specificity. Single point mutations in the Rec domain in a Cas9 from a related species had profound impacts on- and off-target DNA editing, therefore the authors reasoned analogous mutations in GeoCas9 would have similar effects. However, despite a redistribution of local motions and changes in global stability, their chosen mutations had little impact on DNA editing in the context of the full-length enzyme.

In their revised manuscript, the authors were highly responsive to the reviewer's comments incorporating new experimental results including molecular dynamics simulations and RNA binding data using full-length GeoCas9, as well as reframing their discussion and conclusions in consideration of the new data. They were receptive to suggestions for clarification in both the text and methods section. With these changes, the manuscript has been significantly improved.

Their studies highlight the species-specific complexity of interdomain communication and allosteric mechanisms used by these multi-domain endonucleases. The noted strengths of the article remain, and despite the negative results, their approach will garner interest from investigators interested in understanding how the activity and specificity of these enzymes can be engineered to tune activity and limit off-target cleavage by these enzymes. Generally, the manuscript highlights the challenges of studying the effect of allosteric networks on protein function, particularly in multidomain proteins, and thus will be of broad interest to the community.

---

## [Referee Report · Reviewer #3 (Public review)]

The authors explore the role of Rec domains in a thermophilic Cas9 enzyme. They report on the crystal structure of part of the recognition lobe, its dynamics from NMR spin relaxation and relaxation-dispersion data, its interaction mode with guide RNA, and the effect of two single-point mutations hypothesised to enhance specificity. They find that mutations have small effects on Rec domain structure and stability but lead to significant rearrangement of micro- to milli-second dynamics which does not translate into major changes in guide RNA affinity or DNA cleavage specificity, illustrating the inherent tolerance of GeoCas9. The work can be considered as a first step towards understanding motions in GeoCas9 recognition lobe, although no clear hotspots were discovered with potential for future rational design of enhanced Cas9 variants.

Strengths:

- Detailed biophysical and structural investigation, despite a few technical limitations inherent with working with complex targets, provides converging evidence that molecular dynamics embedded in the recognition lobes allow GeoCas9 to operate on a broad range of substrates.

- Since the authors and others have shown that substrate specificity is dictated by equivalent hotspot mutations in other Cas9 variants, we are one step closer to understanding this phenomenon.

Weaknesses:

- Since the mutations investigated here do not significantly affect substrate binding or enzymatic activity, it is difficult to rationalize anything for enzyme engineering at this point.

- Further investigation of the determinants of the observed dynamic modes, and follow-up with rationally designed mutations would hopefully allow to create a real model of the mechanism, but I do understand that this goes beyond the scope of this study.

---

## [Author Response]

The following is the authors’ response to the previous reviews

Responses to final minor critiques following initial revision

Reviewer #1 (Recommendations for the authors):The authors have generally done an excellent job of addressing my and the other reviewers' concerns. I have a few additional concerns that the authors could consider addressing through changes to the text:

We thank the Reviewer for this assessment and are glad to have addressed the major points.

- Regarding the gRNA used for NMR studies, I thank the authors for adding additional rationale for their design of the RNA used. However, I still believe that it is misleading to term this RNA as a "gRNA", given that it is mainly composed of a sequence that is arbitrary (the spacer) and the sections of the gRNA that are constant between all gRNAs are truncated in a way that removes secondary structure that is likely essential for specific contacts with the Rec domains. I do not believe the authors need to make alterations to any of their experiments. However, I do think their description of the "gRNA" should be updated to properly reflect that this RNA lacks any of the secondary structure present in a typical gRNA, much of which is necessary to confer specificity of binding between GeoCas9 and the gRNA. As mentioned in my previous review, this may be best achieved by adding a cartoon of the secondary structure of the full-length gRNA and highlighting the region that was used in the truncated "gRNA".

We understand the Reviewer’s point. For any experiment in which the gRNA was truncated (i.e. NMR or some MST studies), we have clarified the text and no longer call it a “gRNA.” We state initially that it is a portion of the gRNA and then call it simply an “RNA.”

For experiments using the full-length constructs, we have kept the term “gRNA,” as it remains appropriate.

We have also added a final Supplementary figure (S12) showing the structures of the truncated and full-length RNAs used, based on the _Geo_Cas9 cryo-EM structure and predicted with RNAfold.

- Lines 256-257: "The ~3-fold decrease in Kd...". I believe the authors are discussing the Kd's of the mutants relative to WT, in which case the Kd increased. Also, the fold-change appears closer to 2fold than to 3-fold.

Yes, the Reviewer makes a good catch. We have corrected this.

- Lines 407-408: "The mutations also diminished the stability of the full-length GeoCas9 RNP complex." This statement seems at odds with the authors' conclusions in the Results section that the full-length GeoCas9 variants had comparable affinities for the gRNAs (lines 376-382)

We agree that this seems contradictory. In the absence of full-length structures for all variants, we can’t definitively state what causes this. It could be that the mutation has an interesting allosteric effect on structure that does not affect RNA binding but induces the Cas9 protein to simply fall apart at lower temperatures, rendering the binding interaction moot. We have added a statement to this section.

- The authors chose to keep "SpCas9" for consistency with their prior work and the work of many several others, including Doudna et al and Zhang et al. However, I will note that their publications on GeoCas9, the Doudna lab did use SpyCas9 to ensure consistent nomenclature within the publications.

We have made the change to “_Spy_Cas9”

**Reviewer #3 (Recommendations for the authors):**
The authors clearly answered most of my concerns. I still have some technical questions about the analysis of CPMG-RD data but the numbers provided now seem to make sense. While I still think that crystal structures of the point mutant would make the conclusions more "bullet proof", I do appreciate the work associated with this and consider that the manuscript can be published as is.

We agree that additional magnetic fields could allow for additional models of CPMG data fitting and that additional crystal structures of the mutants could add to the conclusions. We appreciate the Reviewer recognizing the balance of the current results and potential future studies in signing off on publication.